# Emerging Role of Deubiquitinating Enzymes (DUBs) in Melanoma Pathogenesis

**DOI:** 10.3390/cancers14143371

**Published:** 2022-07-11

**Authors:** Mickael Ohanna, Pierric Biber, Marcel Deckert

**Affiliations:** 1Université Côte d’Azur, INSERM, C3M, 06204 Nice, France; pierric.biber@univ-cotedazur.fr (P.B.); deckert@unice.fr (M.D.); 2Team MicroCan, Equipe Labellisée Ligue Contre le Cancer, 06204 Nice, France

**Keywords:** melanoma, skin cancer, ubiquitination, deubiquitination, proliferation, invasion, metastasis

## Abstract

**Simple Summary:**

Metastatic melanoma is one of the most aggressive skin tumors with a poor prognosis. Despite the efficacy of immunotherapy and targeted therapies, advanced melanoma patients are often refractory to treatments and have high rates of relapse and death. A major challenge is to identify and understand the mechanisms associated with melanoma development and resistance to gold-standard therapies. Thus, new therapeutic strategies are needed. Ubiquitination is a post-translational modification that plays a crucial role in various cellular biological activities and participates in cancer pathogenesis, including melanoma. Here, we report on the deubiquitination enzymes (DUBs) and their ubiquitin-modified substrates and signaling pathways involved in melanoma progression.

**Abstract:**

Metastatic melanoma is the leading cause of death from skin cancer. Therapies targeting the BRAF oncogenic pathway and immunotherapies show remarkable clinical efficacy. However, these treatments are limited to subgroups of patients and relapse is common. Overall, the majority of patients require additional treatments, justifying the development of new therapeutic strategies. Non-genetic and genetic alterations are considered to be important drivers of cellular adaptation mechanisms to current therapies and disease relapse. Importantly, modification of the overall proteome in response to non-genetic and genetic events supports major cellular changes that are required for the survival, proliferation, and migration of melanoma cells. However, the mechanisms underlying these adaptive responses remain to be investigated. The major contributor to proteome remodeling involves the ubiquitin pathway, ubiquitinating enzymes, and ubiquitin-specific proteases also known as DeUBiquitinases (DUBs). In this review, we summarize the current knowledge regarding the nature and roles of the DUBs recently identified in melanoma progression and therapeutic resistance and discuss their potential as novel sources of vulnerability for melanoma therapy.

## 1. Introduction

### 1.1. Melanoma

Melanoma is a tumor arising from the malignant transformation of melanocytes, pigment-producing cells from the neural crest [1,2]. Several prevalent clinical types of melanoma exist, including cutaneous, uveal, and acral melanoma, according to the location of the transformed melanocyte within the body. In this review, we focus on cutaneous melanoma as it is the most dangerous type of skin cancer, accounting for only 4% of skin cancers, but responsible for approximately 80% of deaths. In the top 20 most prevalent cancers overall, the increasing incidence of cutaneous malignant melanoma accounted for around 330,000 cases (1.6%) of all newly diagnosed cancers worldwide and more than 58,000 deaths in 2021 (Global Cancer Observatory (http://gco.iarc.fr/ (accessed on 1 July 2022)). Melanoma development is due to increased genetic and epigenetic alterations, which create an imbalance in homeostatic signaling pathways. This leads to excessive proliferation of out-of-control tumor cells and subsequent dissemination to distant sites, invading organs and metastasizing. This high mortality and morbidity rate is related to its high ability to metastasize, migrate by tropism, and invade specific sites, such as lymph nodes, the brain, lungs, or liver [3,4]. When metastasis is clinically evident, the prognosis is very poor. Consistent with the Breslow thickness and Clark’s model, melanoma progression is generally described as a linear process, passing first horizontally through the epidermis and then vertically with a high level of proliferative capacity as indicated by the Ki-67 index; these are currently the most important prognostic factors in invasive melanoma [5]. The process is recognized today as being much more complex and less linear in nature. At an early stage, patients with primary cutaneous melanomas are managed by surgical excision with a high remission rate [6] and a good long-term prognosis with a high survival rate at 5 years. Furthermore, the difficulty lies in the diagnosis of early forms of melanoma, with initial management primarily guided by the Breslow thickness (BT), which is the depth of tumor invasion into the dermis, a prognostic method that remains unreliable and demonstrates its shortcomings in risk stratification. There is an urgent need to understand the mechanisms and explore methods of making an accurate assessment of small melanomas with metastatic potential in order to target them and develop new treatments.

Over the past 10 years, new therapeutic options have emerged for the management of patients diagnosed with metastatic melanoma with the discovery of an activating mutation in the gene that codes for the protein kinase BRAF, following studies of the mutational “landscape” in melanoma and supported by more recent data from The Cancer Genome Atlas (TCGA) project [7]. About 50% of melanomas carry activating mutations in the BRAF oncogene and about 30% in the *NRAS* gene leading to overactivation of the mitogen-activated protein kinase (MAPK) pathway, making this signaling cascade a preferential target for melanoma treatment [8]. Several highly selective RAS/MAPK signaling pathway inhibitors have been identified, and this has changed the curative measures applied to these patients. The clinical use of BRAF inhibitors (vemurafenib, dabrafenib, and encorafenib) and MEK inhibitors (trametinib, cobimetinib, and binimetinib) in specific combinations, which have proven to be superior to single-agent therapy, has been shown to be effective in the treatment of melanoma and to significantly improve patients’ progression-free survival and overall survival [9]. Unfortunately, these treatments tend to become progressively less effective, and most patients develop resistance to these inhibitors soon after starting treatment, categorized as acquired resistance, thus adding new categories of patients: those with intrinsic or adaptive (tolerance) resistance to the drugs whose resistance is already present before starting treatment or emerges within hours of treatment. Acquired resistance to inhibitors is commonly caused self-autonomously by genomic rewiring through genetic aberrations of components of the MAPK pathway and its hyperactivation as well as parallel signaling networks such as the PI3K/AKT kinase cascade and alterations in mitochondrial oxidative or redox metabolism [10,11,12,13]. This concept of adaptive capacity is reported in a range of studies as arising from the ability of a sub-population of tumor-derived cells to evolve into a persistent and tolerant state during the initial phases of drug treatment, while most of these tumor cells die [14,15,16]. Single-cell profiling has shown that some of the genetically distinct rare clones, thus fully resistant, can re-enter the cell cycle during treatment and reform the tumor [17,18,19]. The biological events involved are reversible drug or non-genetic adaptation mechanisms that are characterized by changes in the expression of genes involved in cellular plasticity leading to a dedifferentiated state as well as transcriptional, metabolic, and epigenetic signaling pathways [20,21]. Thus, despite the potential of this precision cancer therapy, these treatments also highlight problems of drug resistance that limit the benefit to the patient. Thus, a greater knowledge of the processes of drug adaptation holds the promise of improving the success of melanoma therapy by postponing or reversing acquired resistance.

In recent years, a new era in cancer treatment has emerged with the development of cancer immunotherapy, which seeks to use the patient’s innate and adaptive immune system to recognize and destroy tumor cells. The improved understanding of the immune system based on the modulation of immune checkpoint blocking systems at the cell surface has led to a new age of treatments for melanoma and has become the first-line treatment, even though long-term and durable tumor regression was observed in only a subset of patients [22]. The clinical development of antibodies specifically designed to block immune checkpoint molecules such as CTLA-4 (ipilimumab) and PD-1-/PD-L1-blocking antibodies (pembrolizumab, nivolumab, and atezolizumab) are currently approved as monotherapies for the first-line treatment of advanced melanoma [23,24,25,26]. In spite of such advances, these treatments are also limited by the fact that 40–50% of patients do not respond to these treatments (primary resistance), and, even in responders, resistance to therapy develops in the majority of patients (acquired or secondary resistance). Several causes of resistance to immunotherapy or immune escape have been identified, including defects in antigen presentation in certain tumors or the lack of recognizable foreign antigens. The production of a range of immunosuppressive proteins and metabolic changes both in tumor and T cells are part of the resistance to immune checkpoint blockade therapies [27,28,29].

Accordingly, there is still an unmet need to find alternative therapy options to improve the treatment of melanoma. In this perspective, whether genetic or non-genetic alterations are involved in resistance mechanisms or intrinsic or exogenic drivers leading to epigenetic and transcriptional rearrangement, melanoma cells must fine-tune protein homeostasis and function to support unrestricted cell proliferation. Modulation of the machinery controlling protein ubiquitination, a type of post-translational modification (PTM), has an impact on the stability and functional activity of proteins implicated in a plethora of regulatory pathways such as DNA damage and repair, cell cycle progression, apoptosis, endocytosis, and signal transduction essential for supporting cellular functions. In this review, we focus on recent studies regarding the enzymes involved in protein deubiquitination, the deubiquitinating enzymes (DUBs), which affect the hallmarks of cancer and have a possible impact on melanoma pathogenesis.

### 1.2. The Ubiquitin Pathway

The modulation of cell signaling depends critically on a repertoire of protein posttranslational modification (PTM) mechanisms, which provide an extra regulatory layer that contributes to the functional diversity of the proteome. Protein ubiquitination has emerged as a modification used by signaling processes to regulate a range of functional behaviors [30,31]. Protein ubiquitination (or ubiquitylation) is the dynamic process of covalent binding of the C-terminal glycine of ubiquitin, a small protein of 76 amino acids, to a lysine moiety on protein substrates, whereby serine, threonine, cysteine, and N-terminal methionine moieties can also be modified [32]. Target proteins can be either monoubiquitinated by the addition of a single ubiquitin molecule or polyubiquitinated by the consecutive addition of several ubiquitins to the previous ubiquitin leading to disparate fates of the modified proteins. The designation of the polyubiquitination chain depends on the type of lysine residue (seven lysine residues: K6, K11, K27, K29, K33, K48, and K63) to which the ubiquitin attaches, which also gives rise to a variety of biological outcomes. Lys48-linked chains mainly tag proteins for 26S proteasome-mediated recognition and degradation while K63-related chains play a variety of non-degradative roles and can alter signaling and transcriptional processes as well as protein interaction or localization [30,31]. Monoubiquitination plays an active role in histone regulation and DNA damage repair, signal transduction, trafficking of receptors, and stress response [33,34]. The ubiquitin–proteasome system (UPS) is implicated in the degradation of more than 80% of short-lived proteins in cells and ensures the elimination of useless, damaged, misfolded, and potentially dangerous proteins and the recycle of ubiquitin. Most of the proteins involved in the cell cycle, cell adhesion, migration, invasion, apoptosis, differentiation, angiogenesis and tumor growth, antigen processing, cytokine signaling, transcription, and DNA damage response are regulated by UPS (Figure 1).

The classical cascade of ubiquitin conjugation to a protein substrate (ubiquitination) is initiated by a family of ATP-dependent enzymes called E1-activating enzymes, in which ubiquitin is transferred to the cysteine residue of the active site of E1 with an adenylation of the second ubiquitin, subsequently followed by the transfer of the adenylated ubiquitin to the active site of the E2 ubiquitin conjugating-enzymes (E2 conjugators) and completed by the ligation of ubiquitin to the lysine residues of the target proteins by E3 ligases, which plays a key role in the specific type of ubiquitinated substrate and its associated function [35].

Since ubiquitination is a dynamic and reversible process, the removal of ubiquitin is catalyzed by DUBs. Thus, the main function of DUBs, beyond their role in protein stabilization, is to adjust the degree of protein ubiquitination/deubiquitination, protein activity, and subcellular localization, and to preserve the cellular pool of monoubiquitin. Around 100 DUBs are encoded in the human genome and have been divided into six families based on sequence and structure, including UCH (ubiquitin C-terminal hydrolase), USP (ubiquitin-specific protease), OTU (ovarian tumor proteases), Josephin (Machado-Joseph disease, MJD), ZUP1 (zinc finger-containing ubiquitin peptidase), and the JAMMs (JAB1/MPN/Mov34 metalloenzyme) [36,37,38]. The first five families are cysteine proteases, while the JAMM proteins belong to the zinc-dependent metalloproteinase family. There are approximately 57 USPs, 4 UCHs, 15 OTUs, 4 MJDs, 1 ZUP1, and 9 JAMMs (Figure 2). Due to the number and variety of their substrates, DUBs can drive various cellular processes such as the cell cycle, apoptosis, gene transcription, and DNA repair, and they can exhibit versatile functions in tumor progression, such as epithelial-mesenchymal transition (EMT), cancer stem cell development, metastasis, and tumor microenvironment cross-talk. Here, we compile evidence regarding the involvement of DUBs and related substrates in several biological processes and their relevance in melanoma progression and therapeutic response (Figure 3).

## 2. Deubiquitinating Enzymes in Melanoma

### 2.1. Ubiquitin-Specific Proteases (USPs)

#### 2.1.1. USP4

The ubiquitin-specific protease USP4 is another novel oncogene that belongs to the USP family. It causes a strong increase in melanoma tissue and is almost undetectable in nevus tissue or primary tissue. Deletion of *USP4* does not directly impact melanoma cell proliferation but increases melanoma susceptibility to DNA damage-induced cell apoptosis, depending on p53 signaling [39]. Previous reports have associated DNA repair, p53 stability, and USP4 with cancer [40]. In a large library screening of USP cDNA expression after DNA damage, USP4 was found to be one of a group of peptidases having a profound negative effect on p53 activity, by a degradation-dependent mechanism without affecting its transcription. This regulation depends on the binding of USP4 to ARF-BP1 through deubiquitination, promoting ARF-BP1-dependent ubiquitination and degradation of p53, which indicates its tumor-promoting role [40]. Altered USP4 expression inhibits the invasion and migration capacity of melanoma cells with a drastic decrease in N-cadherin and an increase in E-cadherin mRNA and protein, both markers of epithelial–mesenchymal transition (EMT) [39]. More importantly, USP4 appears to be a regulator of different cellular pathways and targets a variety of substrates. Like the previously described deubiquitinases, USP4 plays a role in critical signaling pathways. It inactivates the NF-κB signaling pathway by targeting TRAF2 and TRAF6 for deubiquitination or by regulating DNA damage response (DDR) with a direct interaction with the MRN (MRE11-RAD50-NBS1) and CtIP complexes that are required for initiating DSB repair [41].

Interestingly, USP4 is one of the DUBs that undergoes activity-dependent relocation. The activated protein kinase B (AKT)-dependent phosphorylated form of USP4 (Ser 445) is an indispensable process that leads to the relocation of USP4 from the nucleus to the cytoplasm where it subsequently exercises its deubiquitinating capacity [42]. These USP4-associated pathways are vital in the pathogenic role of melanoma; however, they remain to be explored.

#### 2.1.2. USP5

It is noteworthy that, among a large number of DUBs, USP5 was the first enzyme studied in relation to the BRAF signaling pathway [43]. When profiling the DUB active site using the HA-UB-VS probe, the DUB catalytic site of USP5 was found to be both downregulated after BRAFinh (vemurafenib) or inversely upregulated after BRAF activation in a heterologous cell line under BRAFV600E overexpression. USP5 has been implicated in the tumorigenesis of various types of human cancers. Consistent with this observation, the depletion of *USP5* in BRAF mutant or non-mutant melanoma cells has a greater effect on growth in 3D models than in 2D ones. Furthermore, overexpression of *USP5* promotes a proliferative advantage in 2D models, but its effect on proliferation-independent anchoring was not proposed in this study. Several DUBs have been reported to regulate p53 stabilization directly or indirectly by suppressing their level of polyubiquitination linked to p53 K48, and USP5 is one of the DUBs involved in these processes. Thus, arrested melanoma cells lacking USP5 are associated with upregulation of p53, p21, and, specifically, p73 in p53 mutant cells, resulting in impaired entry into the G2/M phase of the cell cycle. Furthermore, USP5 deprivation is associated with an increase in p53, p21, and p73 activity. Furthermore, USP5 deprivation leads to an increased sensitivity to the cell death program with increased p53 and caspase 8 and caspase 3 cleavage in combination with BRAFinh [43]. Several USP5 inhibitors have been proposed for human cancers, such as PYR-41, formononetin [44], and WP1130 or one of its optimized analogues EOAI3402143 (G9) [45,46], but these compounds do not exclusively target USP5. In melanoma, G9 synergizes with a BRAF inhibitor and an MEK inhibitor to enhance apoptosis signaling, increase sensitivity to targeted therapies, reverse resistance to vemurafenib in vitro, and reduce tumor growth in vivo [43]. Targeting USP5 offers a potential therapeutic strategy for p53-related melanoma, and, in combination with current therapeutic strategies, it alters cell growth and cell cycle distribution associated with p21 induction in melanoma cells.

#### 2.1.3. USP7

USP7, also known as herpesvirus-associated ubiquitin-specific protease (HAUSP), was originally implicated in many cellular functions such as the cell cycle, apoptosis, and DNA repair through its interaction and stabilization with P53, a tumor suppressor [47]. As USP7 acts on a broader range of chromatin-associated events such as transcription, chromatin organization, and DNA repair through deubiquitination of the histone H2A and H2B variants, it has been found to be a key component in the regulation of DNA repair. The dynamic regulation of H2A or H2B ubiquitination has been associated with several disease processes, particularly in melanoma [48,49]. *USP7* depletion or pharmacological inhibition by P22077 treatment in melanoma or murine (B16) cell lines has been shown to alter tumor expansion in vivo, cell proliferation in vitro, and migration and invasion [49,50,51]. The cytotoxic effect of inhibiting USP7 activity induces ROS production and DNA damage, a mechanism that is independent of p53 signaling. Cells lacking *USP7* show an increased susceptibility to BRAF [49] inhibitors, which makes this DUB a good therapeutic target in melanoma in combination with existing treatments. In tumor progression, consistent findings show that *USP7* levels are increased in both the transition from nevi and benign to cutaneous melanoma and also in melanoma tissue relative to normal tissue [49,50]. In terms of its mode of action, USP7 appears to stabilize and interact with EZH2, which is also implicated in the chromatin repressive complex 2 (PRC2) that mediates transcriptional repression by impairing the transcription of Forkhead Box O1 (FOXO1), the transcription factor that drives genes involved in the apoptotic response, cell cycle, and cell metabolism [49]. The levels of USP7, EZH2, and FOXO1 were found to be correlated with tumor histological grades. Of interest, the proteome of cells lacking *USP7* shows a metabolic pathway dysfunction including an inhibition of the PI3K/AKT/FOXO signaling pathway [50]. Therefore, USP7 plays a role at several scales in melanoma by monitoring the mitogenic activation of the PI3K-AKT signaling pathway, metabolic activation by controlling the redox status of cells, and epigenetic and transcriptional regulation.

#### 2.1.4. USP9X

Another important drug target in melanoma is the deubiquitinase USP9X (X-linked ubiquitin-specific peptidase 9), which prevents the degradation of ubiquitin-specific proteins that are essential in various biological pathways involved in the regulation of cell transformation and survival. USP9X activity and expression were found to be elevated in metastases compared to those in the primary tumor [52]. Thus, enforced expression of *USP9X* enhances tumor growth in vitro and in vivo [52,53]. Particularly for USP9X, its expression and activity are upregulated after ectopic expression of *BRAFV600E* in 293T cells; conversely, inhibition of the kinase partly suppresses USP9X activity in vemurafenib-sensitive but not resistant cells, indicating that USP9X is one of the first DUBs whose activity is dependent on MAPK signaling [52]. Although depletion of *USP9X* reduces tumor growth in melanoma cells carrying *BRAF* or *NRAS* mutations, its combination with MAPK pathway inhibitors (vemurafenib or MEKi) has been shown to enhance the therapeutic response [52]. This improved apoptotic response is mediated by the stability of the transcription factor SOX2, a treatment-induced protein, which, upon depletion of *USP9X*, leads to *SOX2* breakdown, thereby inducing death signals that inhibit growth in vitro and in vivo [52,53]. This USP9X regulation of SOX2 is a recurrent mechanism in prostate cancer and osteosarcoma [46,54]. Of interest, inhibition of USP9X activity by the small-molecule inhibitor DUB G9 also appears to have susceptibility to tumor growth in *NRAS* mutant lines compared to *BRAF* lines in vivo and in vitro. Upon investigation, the deletion or pharmacological inhibition of USP9X was found to reduce the level of *NRAS*. This regulation could be achieved through a direct interaction of USP9X with the transcription factor ETS-1, preventing its proteasomal degradation by deubiquitination, and favoring its binding activity on NRAS promoters in melanoma cells [53]. The ETS family function has been poorly studied in melanoma, with only ETS1 described as favoring invasion and being involved in resistance to MAPKinh. While ETS-1 and ETS-2 are both targets of ubiquitination, loss of USP9X only affects the ETS-1 protein. USP9X inhibition has emerged as a therapeutic strategy alone or in concert with existing therapeutics.

#### 2.1.5. USP13

The deubiquitinase USP13 is a member of the USP subclass of the deubiquitinating enzyme superfamily. USP13 can remove ubiquitin chains from its substrates to inhibit protein degradation. It has been reported that among the DUB family involved in MITF (microphthalmia-associated transcription factor-M) deubiquitination, only USP13 stabilizes and increases the protein levels of MITF, which, in turn, affects the expression of its target gene expression Trpm1 and c-Met [55]. Depletion of *USP13* in MITF-positive melanoma cells reduces MITF protein levels without affecting *MITF* mRNA levels through regulation of its ubiquitination state and the subsequent increase in MITF proteasome-mediated degradation. Consequently, USP13 is involved in cell cycle progression, cell colony formation in vitro, and the growth of tumor xenografts in vivo. This is the only study that implicates USP13 deubiquitinase in melanoma development by regulating MITF protein stability. However, given the critical role of MITF in so many aspects of the developmental and therapeutic response to BRAFinh, the possibility of targeting USP13 activity as responsible for MITF stability needs to be further studied in some aspects of melanoma therapy.

#### 2.1.6. USP14

The deubiquitinating enzyme ubiquitin-specific protease 14 (USP14) represents the only DUB USP family that reversibly associates with the proteasomal 19S regulatory particle [56,57]. This association with the proteasome subunit occurs through the ubiquitin-like domain (Ubl) at the N-terminus of USP14 [58]. Through its strategic location, USP14 serves as a negative and selective regulator of the 26S proteasome, acting as a brake on protein degradation by deubiquitination and facilitating free ubiquitin recycling [58]. The fact that USP14, as well as PSMD14 and UCHL5, are specific subunits of the proteasome makes them the last DUBs encountered prior to proteasome engagement and protein degradation and also makes them attractive therapeutic targets for the treatment of cancers that depend on their ability to absorb the excessive accumulation of toxic proteins and oncogenes. Recent research on melanoma has revealed that USP14 activity inhibition, via the small molecule b-AP15, or loss of function triggers the accumulation of poly-ubiquitinated proteins and a stress response in the ER leading to dysfunctional cell proliferation combined with cell death irrespective of caspase activities [59]. Interestingly, the antitumor effect of USP14 targeting is applicable to *BRAFV600E-*, *NRAS-*, and *NF1*-mutated melanoma cells and to BRAF/MEK inhibitor-resistant cells [59]. Similar observations were made in other cancer types such as myeloma cells [60]. A reduction in tumor growth and the induction of cell death in vivo were observed following b-AP15 treatment in melanoma xenografted mice [59]. While high expression levels of *USP14* are closely correlated with poor prognosis and follow tumor disease progression, its activity was increased in tumor cells as compared to normal melanocyte cells and in BRAFi-resistant cells as compared to the parental cells, resulting in higher responsiveness to b-AP15. These results point to USP14 as a valuable candidate for melanoma treatment and the prevention of resistance to MAPK-targeting therapies.

#### 2.1.7. USP15

USP15 (ubiquitin-specific peptidase 15 or ubiquitin carboxyl-terminal hydrolase 15) belongs to the USP family, and melanoma is one of the cancer tissues where USP15 expression shows a high degree of immunoreactivity (www.proteinatlas.org, accessed on 7 June 2022). The USP15 gene is unusually composed of 22 exons for which eight human isoforms have been described, which may imply specific functions for those isoforms in melanoma. To date, USP15 is an attractive DUB target because of its involvement in both tumor growth and immunity processes [61,62]. Murine or human melanoma cells lacking USP15 were found to exhibit a proliferation arrest and an apoptotic induction in vitro with reduced tumor growth in nude mouse xenografts in vivo. USP15 functionally interacts with and stabilizes the oncoprotein (E3 ubiquitin ligase) MDM2 to avoid its proteasomal degradation and regulates the signaling of p53, a mediator of cell survival. This mechanism appears to be maintained in other cancers such as colon cancer [61] and glial cancer [63], as well as in lymphocyte cell sub-populations such as T cells, but not in murine B cells or thymocytes nor in human primary fibroblasts [61]. Evidence regarding the relationship between USP15 and the melanoma immune response is based on two investigations [61,62]. In *Usp15* −/− (KO) mice as compared to wild-type (WT) mice and after B16 injection, the growth curve of melanoma cells was profoundly reduced with a prolonged lifespan. In such mice, *Usp15* deficiency enhanced MDM2/NFATC2 pathway-dependent T-cell activation in vitro and a strong T-cell tumor infiltration in vivo with a major contribution of T-cell-mediated IFN-γ within the anti-tumor immune response [61]. The underlying mechanism behind this control places the USP15 axis and the immune response in relation to epigenetic modifications, particularly of the enzymes that catalyze DNA methylation in the ten-to-eleven translocation (TET) family, enzymes frequently altered in hematopoietic pathologies. Therefore, USP15 interacts with and inactivates TET2 by altering its DNA binding to the inflammatory genes. In contrast, the deletion of USP15 enhances TET activity and tumor-intrinsic chemokines as well as infiltration and IFN-γ production from active T cells. Further studies providing additional support for USP15 functions in IFN signaling include IκBα regulation and the E3 ligase TRIM25, which both play a critical role in the innate and inflammatory immune response [64,65]. At this stage, USP15 in melanoma appears to be involved in the anti-tumor immune response and its targeting could be a useful strategy to inhibit growth by inducing apoptosis of tumor cells by activating anti-tumor immunity and/or by boosting the efficacy of existing immunotherapy.

#### 2.1.8. USP22

Ubiquitin-specific peptidase 22 (USP22) is a novel DUB in melanoma that has been linked to cell cycle progression, treatment resistance, metastasis, and immune response. USP22 is widely regarded as an oncoprotein; its aberrant expression has been associated with poor cancer prognosis in various types of human cancer, including melanoma patients. A higher expression of USP22 has been observed in metastatic melanoma compared to that in the primary tumor, indicating an important role in melanoma progression [66]. Recently, knockdown of USP22 in reduced melanoma cells has been associated with cell proliferation. In this mechanism, USP22-dependent proliferation is mainly under the dependence of Ye-associated protein 1 (YAP) pathways. In melanoma patient samples, a correlation between USP22 and YAP was observed, and, as a result, high *USP22* expression was associated with high *YAP* expression. In this study, USP22 specifically interacted with and de-ubiquitinated YAP, as well as downstream target genes such as *CTGF* and *CYR61*. Depletion of *USP22* decreased the half-life of YAP protein levels, but not mRNA levels, and significantly suppressed *CTGF* and *CYR61* [2,67]. Recently, YAP/TAZ activation was found to enhance resistance to MAPK inhibitor drugs. Under PLX 4032 treatment, drug-resistant melanoma cells exhibited increased nuclear accumulation of YAP/TAZ and transcriptional activity, which, in turn, conferred resistance to BRAF inhibitors in BRAF V600E mutant melanoma cells [68]. Thus, overexpression of *USP22* resulted in resistance to the BRAF inhibitor vemurafenib through the stabilization of YAP and opened new therapeutic avenues for combined inhibition of USP22/YAP and BRAF as an option for melanoma treatment [67]. Another way to regulate tumor growth is to modulate the potential of tumors to hijack immune recognition and immunological response. It was recently reported that the significant reduction in tumor volume/weight upon stable knockdown of USP22 in B16 is associated with a decrease in PD-L1 stability. Conversely, overexpression of USP22 prolongs the half-life of the PD-L1 protein [69]. This study highlights a novel role for the tumor suppressive properties of USP22 via regulation of PD-L1, which suppressed antitumor immunity in melanoma mouse tumor models but also in other cancers [70]. However, the role of USP22 remains controversial at this stage and was challenged by a study showing that suppression of *Usp22* expression in B16-OVA melanoma cells markedly reduced the immune response. Inversely, overexpression of *Usp22* increased the sensitivity of T-cell-mediated killing. The mechanism of immune therapeutic response dependent on USP22 involves STAT1 stabilization via deubiquitination and promotes the FN-JAK1-STAT1 signal, which is a pathway involved in immune therapy resistance in melanoma [68]. How USP22 senses tumor microenvironment signals to regulate PD-L1, and whether other deubiquitinating enzymes are able to sense these signals, remains unknown. These new findings may provide potential therapeutic targets for enhancing the efficacy of T-cell-based immunotherapy. USP22 controls cell growth and activation in different ways: it induces changes in gene promoter regions by removing ubiquitin moieties from histones H2A and H2B leading to transcription activation [71], and it induces cell cycle progression by stabilizing *TRF1* [72], *COX-2* [73], *CCNB1* (Cyclin-B1) [74], *CCND1* (Cyclin D1), *NFAT*, and *SIRT1* [75], which regulate the genes involved in metabolism, cell cycle, invasion, and apoptosis, a pathway implicated in melanoma progression [76]. However, the functional impact of USP22 deubiquitylation of histone or non-histone substrate regulation has not yet been studied in the melanoma process.

#### 2.1.9. USP28

As a nucleoplasm-located DUB, USP28, ubiquitin-specific peptidase 28, is among the deubiquitinases that alter the stability and turnover of critical cancer oncoproteins such as cMYC involved in the proliferation and aggressiveness of colon and breast carcinoma and glioblastoma cells [77,78,79]. Recently, efforts have been made to investigate DUBs involved in MAPK signaling pathway modulation. Upon functional shRNA screening of DUBs, USP28 was recognized as a regulator of the stability and abundance of BRAF, a major proto-oncogene driver of MAPK cascade upregulation in cell culture models and in vivo [80]. Although lack of *USP28* expression in melanoma cells has no impact on cell proliferation, it favors both the development of resistance to BRAF/MEK inhibitor combination therapies in vitro and in vivo and the generation of emergent tumor cells in mouse xenograft experiments. This short-term resistance results in an impairment of apoptosis markers in cells lacking USP28 under treatment. Of clinical relevance, decreased USP28 expression is more frequently observed in *BRAFV600E* melanoma patients and leads to a poorer overall survival, indicating the possible role of USP28 as a relevant factor in melanoma progression. Moreover, USP28, in addition to its role in tumor-promoting substrate stabilization qualifying as a proto-oncoprotein, it also controls the stability of several proteins involved in DNA damage response (CHEK2) [81], the metabolism (HIF1) [82], translation (JAK-STAT) [83], and the cell cycle (TP53) [84], all of which are relevant pathways in melanoma development and MAPKinh response.

#### 2.1.10. CYLD

CYLD (cylindromatosis-associated DUB) belongs to the USP class of cysteine proteases and is identified as a tumor suppressor on account of its loss of function correlated with several cancers including melanoma. CYLD expression was shown to be inversely correlated with overall and progression-free survival in melanoma patients [50,85,86]. Initial investigations into the role of CYLD in malignant melanoma have shown that its transcript or protein level is strongly decreased in primary or metastatic melanomas compared to normal human epidermal melanocytes in vitro as well as in vivo on immunohistochemistry sections [85,87]. Furthermore, *CYLD* expression is not influenced by either origin or genetic background and remains highly heterogeneous both in melanoma cells and in fresh tissue melanoma progression [86]. However, CYLD has been implicated in the control of melanoma tumorigenesis in vivo. Melanoma development and progression were studied in *Cyld*-epleted mice (C57BL/6 Cyld knockout) in a mouse model of spontaneous melanoma development (Tg (Grm1) Cyld +/+ mice). These *Cyld*-deficient mice showed an earlier onset and accelerated growth of melanoma compared to wild-type mice due to lymphatic angiogenesis caused by its deubiquitinase function. However, this process is a tumor-autonomous self-mechanism, as cell lines generated from these mice (Tg(Grm1) Cyld −/−) showed increased proliferation and migration, as well as clonogenicity in vitro [88]. Thus, the deubiquitinase function of *CYLD* appears to be more relevant to vascular remodeling than to the proliferation process. *CYLD* expression is mediated by the transcriptional repressive *SNAIL1* and by promoter regulation binding dependent on ERK signaling. Thus, a decrease in *SNAIL1* expression after treatment with a BRAF inhibitor leads to an increase in *CYLD* mRNA levels and, conversely, after transfection with a mutated form of *BRAF*. Melanoma cells overexpressing *CYLD* showed an anchorage-independent reduction in proliferation and growth demonstrated both in vivo and in vitro using a murine xenograft model. Other studies on in vitro or in vivo melanoma have also concluded that this process is dependent on deubiquitinase activity [86,87]. Conversely, *CYLD* expression has been suggested to be downregulated because of elevated *SNAIL1* expression, resulting in increased levels of *CCND1*(Cyclin-D1) and *CDH2* (N-Cadherin) and enhanced proliferation, migration, and invasion of human melanomas [85]. Additional investigations have shown that this control of proliferation after *CYLD* depletion is due to the induction of apoptotic markers such as activation of caspase 3 and cleavage of poly(ADP-ribose) polymerase (PARP). This mechanism involves the suppression of the K63-linked polyubiquitin chains of RIPK1 (receptor-interacting protein kinase 1) by CYLD, thereby promoting its degradation, a mechanism that does not appear to be exclusive to melanoma [86]. RIPK1 overexpression is known to promote caspase activation-dependent cell death in many other cell types [89]. Thus, depleting *CYLD* in melanoma cells leads to an increase in RIPK1 associated with an increase in K63-related polyubiquitination of RIPK1. At the molecular level, some mechanisms are maintained, such as activation of NF-κB and JNK/AP-1/β1-integrin pathways in response to *CYLD* gain and loss of function that could explain the observed melanoma tumorigenesis [87,90]. Recently, CYLD has been shown to participate in the tumorigenicity of melanoma cells through its important role in regulating the MAPK pathway by modifying lysine-63 (K63)-linked polyubiquitinated chains for ERK1/2. Forced expression of CYLD results in a hypoubiquitinated state of ERK1/ERK2 with reduced activation, but not observable with a catalytically inactive CYLD mutant (C601A). This inverse correlation between CYLD and ERK1/2 activation is observable by immunohistochemistry on human melanoma samples with a preference for a decreased *CYLD* transcript in highly metastatic melanoma and is associated with poor survival in melanoma patients. This regulation involves an interaction between CYLD and ERK1/2 in A375 cells. In addition, melanoma-derived point mutants of CYLD (F675S, P698L, and D681G) also lose the ability to remove ubiquitin chains from ERK1. For the first time, the ubiquitination role of ERK1 and ERK2 kinases, involved in the major signaling pathways of fundamental processes such as cell proliferation, survival, migration, and differentiation, has been identified in melanoma. K63 ubiquitination and ERK1/2 activation are closely correlated, as the increase in ERK1/2 proteins conjugated with K63 ubiquitin and their phosphorylation were activated after stimulation with growth factors. Among the 11 lysine residues of ERK1, ERK1 ubiquitination occurs mainly at residues K168 and K302, a site maintained on ERK2 that corresponds to K149 and K283. Expression of ERK1 and ERK2 proteins, which are mutant for these ubiquitination sites, results in decreased ERK activation and renders melanoma cells resistant to the BRAF inhibitor PLX4032 (vemurafenib). Moreover, overexpression of these mutants does not cause any cytotoxic effect in A375 cells as does forced expression of ERK1 or ERK2. Interestingly, ubiquitination of ERK2 at the K302 site, by its structural localization, favors interaction with MEK without modifying activation, which makes ubiquitination of ERK2 a drug strategy of choice for the activation of the MAPK pathway that is mostly reactivatable in BRAFinh resistance mechanisms.

### 2.2. JAB1/MPN/MOV34 Metalloenzymes (JAMMs)

#### 2.2.1. MYSM1

During the transformation of cells into cancer cells, overexpression of the enzyme Myb-like SWIRM and MPN domain 1 (MYSM1, also known as 2A-DUB) allows for an increased expression of genes involved in cell growth and proliferation. This functional involvement in MYSM1 genome regulation is due, in part, to chromatin-associated factors and catalyzes the deubiquitination of histone H2A on lysine residue K119. More recently, it has also been implicated in poor outcomes in melanoma [91]. Of interest, MYSDM1 has been identified as involved in the proper functioning of melanogenesis, as well as in melanoma tumorigenesis [92]. Melanoma clones lacking *MYSM1* have been found to exhibit reduced overall proliferation and viability, including in 3D culture models, with increased apoptosis. In addition to the growth defects, MYSM1-deficient mice exhibit “belly and tail patches” (Bst) characteristic of a phenotypic dysfunction of skin development that intensifies with age. These mice show abnormalities in the specification of murine melanocytes, which affects the tyrosinase (Tyr) genes and causes the loss of epidermal stem cells implicated in derived precursor regeneration. These processes are dependent on the P53 pathway and the expression of the cMET transcription factor gene through regulatory *MYSM1*-binding to promotor elements. An MYSM1 protein quantification showed an incremental increase during the progressive transformation of normal melanocytes into malignant melanoma cells with a correlation with c-KIT expression in melanoma. *MYSM1* expression is sensitive to chemotherapy-induced DNA damage response (etoposide) with nuclear recruitment to the γH2AX site in melanoma cells [93]. The therapeutic potential of MYSM1 in the maintenance of genome integrity could be explored by targeting the ubiquitinated form of its substrate histone H2A (H2Aub), as well as other components of the MYSM1 interactome, including DNA repair and replication factors, previously implicated in DNA damage responses and melanoma cell survival [92,93].

#### 2.2.2. PSMD14

PSMD14 (RPN11 or POH1) is a member of the JAMM domain class of metalloproteinases with ATP- and Zn^2+^-dependent deubiquitinase activity. This domain, composed of two histidine residues and aspartic acid (His-X-His-X10-Asp), allows binding to a zinc ion to form the catalytic site of PSMD14. This DUB is an intrinsic component of the 26S proteasome that sits directly above the substrate translocation gate in the 19S RP lid [94]. It is a strategically important location for the control of substrate recognition prior to degradation, thus acting as a gatekeeper for ubiquitinated substrate entry and participating in the release of ubiquitin chains after cleavage. Studies have shown the importance of the JAMM motif of PSMD14 in the deubiquitination of substrates by the proteasome [95]. Thus, PSMD14 may have a dual function after the removal of the ubiquitin chains, either facilitating the delivery of the substrate to the proteasome and facilitating its proteolysis, or conversely releasing the protein from the proteasome and preventing its degradation. PSMD14 exerts both a proteasome-dependent deubiquitinase activity via cleavage of K48-linked ubiquitin chains for proteasomal processing and degradation, and a non-proteasomal activity by targeting K63-linked ubiquitin chains [96,97,98]. As suspected, due to its key role in proteasomal integrity and activity, PSMD14 functional alteration leads to cell viability defects in all species and has recently been studied in melanoma. After a screening of the DUB siRNA library, depletion of *PSMD14* was found to be the main process in both the suppression of growth by controlling SMAD3 stabilization and activation and the suppression of metastatic capacity by affecting the TGFb pathway and the transcription factor SLUG [99]. In addition to melanoma cellular proliferation, these findings suggest a role of PSMD14 within the EMT process, as also observed in breast and esophageal cancers [100,101]. Hence, PSMD14 activity or expression may mediate the EMT phenotype that is associated with tumor progression, metastasis, and drug resistance [102,103,104]. Recently, capzimine has been developed as a selective inhibitor of PSMD14. Capzimine induces cell death, is associated with DNA damage, and blocks proliferation in 60 cancer cell lines (NCI panel), including melanoma cell lines [105]. At this point, PSMD14 appears to be a promising new therapeutic target in melanoma. However, further studies are required to better understand the cellular and molecular mechanisms involving PSMD14 in relation to either degradative or non-degradative processes. Regarding the role of PSMD14 in cell survival, additional research should also address the potential of targeting PSMD14 in combination with current anti-MAPK therapies or immune checkpoint blockade therapies.

## 3. Conclusions and Perspectives

Despite recent advances in melanoma treatment, with the use of therapies targeting BRAF oncogenic signaling and anti-PD-1 immunotherapies, treatment failure and disease relapse occur in a majority of melanoma patients with advanced late-stage melanoma, warranting the development of novel therapeutic options. This review article focuses on the role of DUBs and their ubiquitin-modified substrates in the pathogenesis of cutaneous melanoma at the level of tumor initiation, survival/proliferation, and migration/invasion, processes that drive tumor progression. Among the DUBs encoded in the human genome, twelve are experimentally involved in melanoma progression through their alteration and are related to a poor prognosis in melanoma patients. In addition to USP10, USP11, and USP22, which are associated with a more aggressive and invasive phenotype linked to clinico-pathological parameters, USP9X, USP7, CYLD, and MYSM1 are also implicated in the transformation of melanocytes to melanoma and in the melanoma metastatic processes [66]. These DUBs are involved in a variety of biological processes, including proteasomal and transcriptional programs, DNA damage response, inflammation, and MAPK oncogenic signaling pathways that may lead to therapeutic resistance. In this way, these DUBs could be considered as emerging biomarkers by which to predict metastatic potential and response to treatment. Additional research may address functional relationships between the involvement of deubiquitinase in both genetic alterations that contribute to the molecular classification of melanoma [106] and non-genetic alterations (metabolism and epigenetic changes) implicated in key signaling pathways responsible for melanoma progression and resistance [107,108].

Regarding the role of PSMD14 in cell survival, additional research should also address the potential of targeting PSMD14 in combination with current anti-MAPK therapies or immune checkpoint blockade therapies. Therefore, investigating the effect of compounds targeting these DUBs in combination therapy regimens may pave the way for future therapies for metastatic and resistant melanoma. For example, USP7 has recently gained attention in cancer biology due to its central role in stabilizing the tumor suppressor p53. USP7 activity has been successfully targeted with small-molecule inhibitors FT827 and FT671, which showed cytotoxic effects in in vitro and in vivo models [47,109]. At present, these pharmacologic inhibitors have not been assessed in clinical trials. A clinical trial testing the USP14 and UCHL5 competitive inhibitor VLX1570 in combination with dexamethasone in patients with multiple myeloma (NCT02372240) was suspended due to high toxicity. In this context, testing lower doses of VLX1570 (or related compounds) in combination with current anti-melanoma therapies may reveal novel therapeutic opportunities. In addition, PSMD14 inhibitors may provide an alternative to current proteasome inhibitors due to a better efficacy on solid tumors, particularly melanoma cells [105]. To date, however, most of the small-molecule DUB inhibitors developed suffer from poor potency and selectivity. Thus, the non-selective quality of some DUBs inhibitors is highlighted by two main barriers: the first barrier that has impeded the development of drugs targeting DUBs is the similarities in catalytic domains among each subclass. This has rendered it difficult to find molecules that show selectivity among related DUBs without having off-target effects. For instance, UCHL5 and USP14, the proteasomal DUBs, share similar active sites, so therapy specific to a single DUB remains to be improved. A second barrier relates to the discovery that many DUBs have several substrates, implicated in several protein complexes, molecular pathways, and cellular identities. Thus, a specific inhibitor might have limitations in its mechanism of action [110]. Hence, understanding the mechanisms by which individual DUBs act is important in initiating any subsequent screening and drug discovery campaign. Efforts are being undertaken through large-scale approaches, including a combined computational and experimental approach that can be used to propose potential new DUB functions and provide useful resources for interested investigators [111].

Owing to their biological activity and their druggability, DUBs represent fascinating targets. As such, great advances have been made by attributing specific functions to DUBs in multiple processes related to melanoma pathology. However, before DUB targeting can have clinical application, a better understanding of their mechanisms of action is needed and future studies should address several key issues, such as the following: What are the substrates of these direct DUBs in melanoma? Are they the same as those in other tissues or cancers? How are these DUBS adjusted at these distinct pathological stages? Are melanoma pathological processes regulated by degradative ubiquitination, non-degradative ubiquitination, or a combination of both? Does environmental diversity, such as metabolic changes [112], circulating growth factor levels, changing physical stresses or different angiogenic or inflammatory zones [113,114], or the mutational context as “pilot” [59] or “transient” mutation, affect the degree of deubiquitination, the expression level of DUBs, and the diversity of the substrates of interest in melanoma? Both the modulation of deubiquitinase activity and targeted substrates contribute to this diaphony affecting cellular fate decisions directly relevant to cancer, hinting that the same DUB can be assigned properties of both oncogenes and tumor suppressors depending on the cellular and environmental context [115,116]. Answering these questions should certainly ameliorate the translational potential of targeting DUBs in advanced and refractory melanoma.

## Figures and Tables

**Figure 1 cancers-14-03371-f001:**
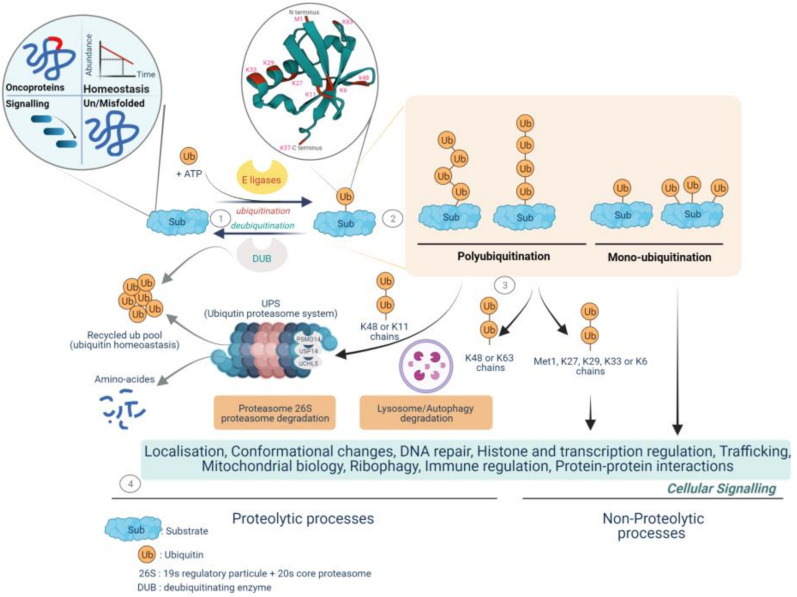
Fate of protein-tethering ubiquitin, a post-translational modification, to proteolytic and non-proteolytic pathways, resulting in specific cellular responses. The ubiquitination process drives protein homeostasis by controlling abundance, temporal and structural integrity, proper localization, and protein non-mutational burden. Called the quality control function, it supports nearly all the cellular functionalities implicated in protein–protein interactions, gene expression, signal transduction cascades, and metabolic pathways. **1**—Protein ubiquitination is performed by the coordinated activity of ubiquitin ligase (E1, E2, and E3 enzymes) and deubiquitinating enzymes (DUBs), called deubiquitination, by antagonizing ligase activity and altering the substrate fate. **2**—Ubiquitin is covalently transferred (isopeptide bonds) between the C-terminal glycine residue (Gly) and substrate lysine residues (Lys) to form monoubiquitinated proteins or can join up with other ubiquitin molecules at the intrinsic N-terminal Met1 residue and/or at the seven intrinsic Lys residues (Lys6, Lys11, Lys27, Lys29, Lys33, Lys48, and Lys63) and may form multi- or poly-ubiquitin chains. **3**—The fate and function of ubiquitinated proteins are affected by the topology and type of ubiquitin- binding. The K48/K11 polyubiquitinated chains have historically been identified as mediating proteasomal degradation of normally folded short-lived proteins and recycling ubiquitin called the ubiquitin–proteasome system (UPS). To date, three deubiquitinases, the metalloprotease PSMD14 and the two cysteine proteases UCHL5 and USP14, have been found to be components of the proteasome 19S regulatory particle implicated in both binding of the ubiquitinated substrate and proteasome activity. **4**—The UPS is a selective and irreversible protein removal mechanism that controls signal transduction, cell division, stress response, and immune adaptation. The degradation of misfolded proteins by the UPS or autophagy is mainly mediated through K48/K63 branch chains. Met1/K63/K29 linkage can modulate various non-degradative processes such as signal transduction, DNA repair, and kinase activation. Lys6-linked chains have been identified as being involved in mitophagy. Monoubiquitination plays various roles in such functions as protein trafficking, DNA repair, chromatin remodeling, and regulation of transcription. (Created with BioRender.com, accessed on 20 June 2022).

**Figure 2 cancers-14-03371-f002:**
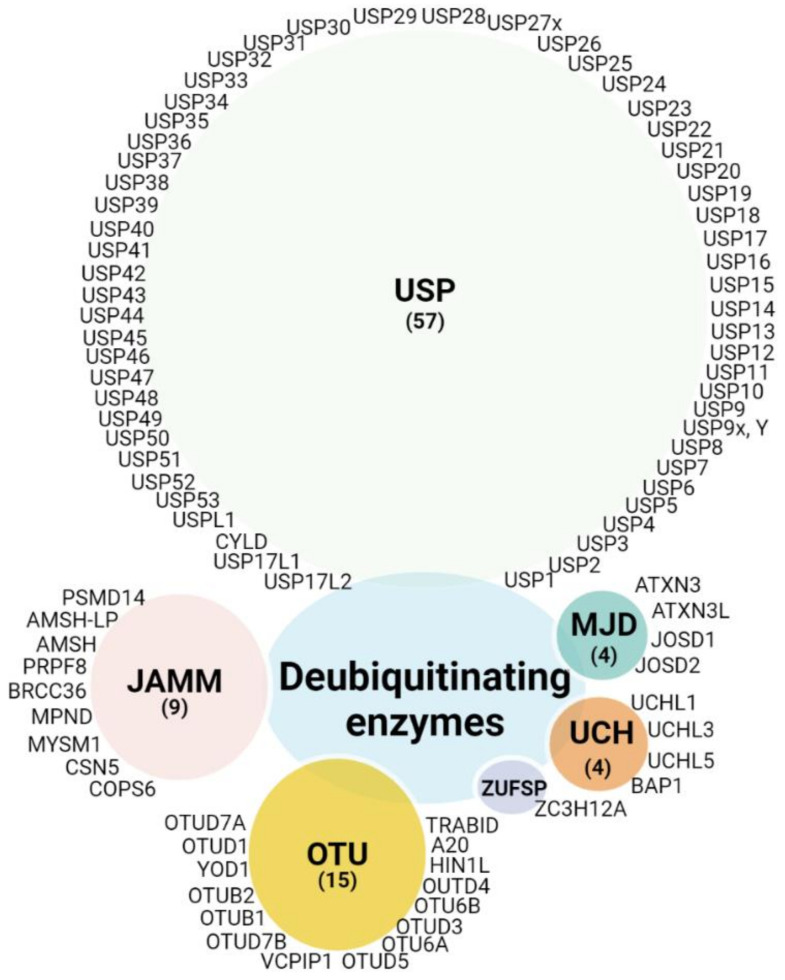
DUB categories: DUBs currently encoded in the human genome are clustered in five classes depicted in different color schemes. Of the six subfamilies, five are cysteine proteases: ubiquitin C-terminal hydrolases (UCH), ubiquitin-specific proteases (USP), zinc finger-containing ubiquitin peptidase (ZUP1), Machado-Joseph disease proteases (MJD, Josephins), and ovarian tumor proteases (OTU), and a one family belongs to the Jab1/Pab1/MPN domain-associated zinc metalloproteases (JAMM). (Created with BioRender.com, (Created with BioRender.com, accessed 20 June 2022).

**Figure 3 cancers-14-03371-f003:**
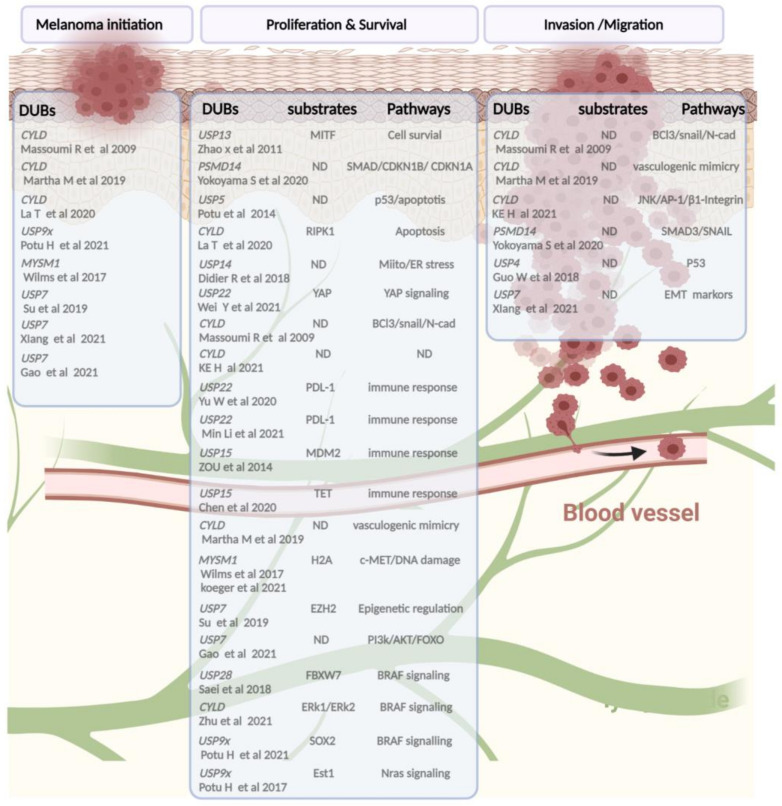
A schematic snapshot of deubiquitinases involved in melanoma pathogenicity based on published studies. From left to right, the following categories are shown: DUBs implied in tumor initiation or progression, mainly found in vivo studies, followed by a list of DUBs and their substrates and/or signaling pathways leading (in bold) to alteration of the proliferation, therapeutic response adaptation, and invasion/migration processes of melanoma cells. (Created with BioRender.com, accessed on 20 June 2022).

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
