# Peer review of "Emerging Role of Deubiquitinating Enzymes (DUBs) in Melanoma Pathogenesis"

_cancers, 2022, doi:10.3390/cancers14143371_

Round 1

Reviewer 1 Report

This work is very interesting, but there are a few things that need to be clarified.

Major comments:

11) the nomenclator associated with the spelling of the names of human genes and proteins is not used correctly in the paper 

22)    the authors refer their analysis to the molecular pathways and proteins involved in them, however, they only briefly address the issue of molecular succlassification of melanoma (line 54). I believe that both to understand the role of DUBs and to determine the potential therapeutic use, more extensive reference to the findings in this regard is necessary. Especially since the authors conclude that such aspects have been discussed.

This review article focuses on 576 the role of DUBs and their ubiquitin-modified substrates in the pathogenesis of cutaneous 577 melanoma, at the level of tumour initiation, survival/proliferation and migration/inva- 578 sion, processes that drive tumor progression, and the level of therapeutic response. 579 Among the DUBs encoded in the human genome, twelve are experimentally involved in 580 melanoma progression through their alteration and are related to a poor prognosis in mel- 581 anoma patients.

One of the works in this area is:

Rabbie R et al. Melanoma subtypes: genomic profiles, prognostic molecular markers and therapeutic possibilities. J Pathol2019;247:539–551

The spelling of genes and proteins is properly used in this work.

Minor comments

1) The text is overloaded and difficult to read. In many places, you could combine two sentences and the information they contain and create one. This would make the content conveyed more transparent. There are repetitions of the same phrases / names in close neighborhood, e.g. lines 121 and 123.

2) The description of the Figure 1 needs to be reformatted. Perhaps the solution is to introduce the numbering of steps to the figure in order to divide the legend into shorter parts.

Author Response

We would like to thank the reviewer for these informative remarks. Regarding major comments 1) We have restored official nomenclature for the names of human/mouse genes and proteins in the entire manuscript identified in highlighted in yellow fluorescence. 2) The suggested reference provides an effective insight into the molecular sub-classification of melanoma and furthermore pushes the inquiry on Deubiquitinase may favorably be implicated and shape the different genomic landscape identified. Therefore in Conclusion and Perspectives we have included the remark from line 616 to line 620: “Additional research may address functional relationships between the involvement of Deubiquitinase in both genetic alterations, that contribute to the molecular classification of melanoma [106] and non-genetic alterations (metabolism and epigenetic changes) implicated in key signaling pathways responsible for melanoma progression and resistance [107] [108] »

Regarding minor comments 1) The entire paper was sent to the proposed "Cancers" Edition with a certificate proving the operation. (PDF upload) 2) Figure 1 has been modified as recommended with a numbering clarifying various steps explained in both the text and the iconography.

Reviewer 2 Report

The molecular events promoting the genesis and progression of melanomas warrant close inspection and Ohanna, Biber and Deckert have done a commendable job of distilling the key findings implicating DUBs in melanoma from an increasing formidable literature.  The manuscript is written in a lucid and compelling style and is refreshingly devoid of errors of grammar or typography.  The figures are informative and visually appealing.  This is a very solid document and I have only minor concerns.  The first relates to references.  I understand that in a review of this nature there is a tension between providing references for each claim that has been made and some arbitrary limit imposed on their number (which in this case appears to be 100).   Having faced this conundrum in the past I am sympathetic to their position but if the limit on references cannot be increased (as I believe it should) I believe the distribution of references should be revisited.  As an example no references are provided for any of the numerous assertions made in the section on USP4 and and only one in the subsequent section on USP5, whereas the next two sections (on USP7 and USP9X) each contain a modest but reasonable number.  It is rather jarring to encounter a definitive statement along the lines of "In a large screening of the USP cDNA expression library after DNA damage, only USP4 was reported to induce p53 degradation" (lines 210/211) without the citation being given.  This is a recurrent problem that could be rectified by an increase in the citations allowed (an allowance that would greatly improve the impact of the paper).  My second suggestion would be to provide a brief mention of the primary obstacle that has hindered the development of drugs targeting DUBS: the similarity of catalytic domains within each subclass.  This has made it difficult to find molecules that would target a specific DUB and limit off-target effects.  A second obstacle relates to the finding that many DUBs have multiple substrates (often in different molecular pathways), so even a specific inhibitor might have its limitations.  These are issues that could be mentioned in the Conclusions and Perspectives section.  The final suggestion relates to the recurrent ambiguity evident in the literature for the oncogenic versus tumor suppressive activity of DUBs (these opposing claims have been made for many, if not most DUBs - see PMID: 33415735).  The "double agent" property is discussed in a broader context in PMID: 33396222.  In networked signalling pathways (that invariably feature DUBs) context clearly matters, and the failure of therapies in melanoma that have worked in other cancers may be explained by contextual differences (tumor microenvironment, mutational profile, etc.).

Author Response

Concerning the omission of references, it has effectively been corrected in the section "USP4" (line 224 to 248). In this section, highlighted in red identifies the corrections corresponding to references.  Moreover, Regarding the previous sentence "In a large screening of the USP cDNA expression library after DNA damage, only USP4 was reported to induce p53 degradation" this was removed and changed to a lighter and more appropriate USP4 statement (red color - line 229 to 232). « In a large library screening of USP cDNA expression after DNA damage, USP4 was found to be one of a group of peptidases having a profound negative effect on p53 activity, by a degradation-dependent mechanism without affecting its transcription »

We are in agreement with the reviewer's assessment of the analysis on the second suggestion. Effectively we have included this important insight concerning the difficulty of targeting these enzymes, DUBs, regardless of their structural similarity within each subclass and their biological function. We have added a red comment from line 636 to 650:” “To date, however, most of the small-molecule DUB inhibitors developed suffer from poor potency and selectivity. Thus, the non-selective quality of some DUBs inhibitors is highlighted by two main barriers: the first barrier that has impeded the development of drugs targeting DUBs is the similarities in catalytic domains among each subclass. This has rendered it difficult to find molecules that show selectivity among related DUBs without having off-target effects. For instance, UCHL5 and USP14, the proteasomal DUBs, share similar active sites, so therapy specific to a single DUB remains to be improved. A second barrier relates to the discovery that many DUBs have several substrates, implicated in several protein complexes, molecular pathways, and cellular identities. Thus, a specific inhibitor might have limitations in its mechanism of action [110]. Hence, understanding the mechanisms by which individual DUBs act is important in initiating any subsequent screening and drug discovery campaign. Efforts are being undertaken through large-scale approaches, including a combined computational and experimental approach that can be used to propose potential new DUB functions and provide useful resources for interested investigators [111]. »

In addition we appreciate the reviewer's final suggestion for improving the current manuscript. Hence, we incorporated in the Conclusions/Perspective section the importance of oncogenic versus tumor suppressive activities of deubiquitinating enzymes in relation to melanoma based on the proposed reviews. We have added a blue comment from line 647 to 655. “Does environmental diversity, such as metabolic changes [112], circulating growth factor levels, changing physical stresses or different angiogenic or inflammatory zones [113,114], or the mutational context as 'pilot' [59] or 'transient' mutation, affect the degree of deubiquitination, the expression level of DUBs, and the diversity of the substrates of interest in melanoma? Both the modulation of deubiquitinase activity and targeted substrates contribute to this diaphony affecting cellular fate decisions directly relevant to cancer, hinting that the same DUB can be assigned properties of both oncogenes and tumor suppressors depending on the cellular and environmental context [115,116].”

Round 2

Reviewer 1 Report

Manuscript has been sufficiently improved to warrant publication in Cancers.